# Clinical Evidence behind Stereotactic Radiotherapy for the Treatment of Ventricular Tachycardia (STAR)—A Comprehensive Review

**DOI:** 10.3390/jcm10061238

**Published:** 2021-03-17

**Authors:** Marcin Miszczyk, Tomasz Jadczyk, Krzysztof Gołba, Wojciech Wojakowski, Krystian Wita, Jacek Bednarek, Sławomir Blamek

**Affiliations:** 1IIIrd Department of Radiotherapy and Chemotherapy, Maria Sklodowska-Curie National Research Institute of Oncology, 44-102 Gliwice, Poland; 2Department of Cardiology and Structural Heart Diseases, Medical University of Silesia, 40-055 Katowice, Poland; tomasz.jadczyk@gmail.com (T.J.); wojtek.wojakowski@gmail.com (W.W.); 3International Clinical Research Center, Interventional Cardiac Electrophysiology Group, St. Anne’s University Hospital Brno, 664/53 Brno, Czech Republic; 4Upper-Silesian Heart Center, Department of Electrocardiology, 40-055 Katowice, Poland; krzysztof.golba@gmail.com; 5Department of Electrocardiology and Heart Failure, Medical University of Silesia, 40-055 Katowice, Poland; 6First Department of Cardiology, Medical University of Silesia, 40-055 Katowice, Poland; welwetek@poczta.onet.pl; 7Department of Electrocardiology, John Paul II Hospital, 31-202 Cracow, Poland; bednareks@op.pl; 8Department of Radiotherapy, Maria Sklodowska-Curie National Research Institute of Oncology, 44-102 Gliwice, Poland; slawomir.blamek@io.gliwice.pl

**Keywords:** ablation, noninvasive, radiosurgery, stereotactic body radiation therapy, substrate ablation, ventricular tachycardia

## Abstract

The electrophysiology-guided noninvasive cardiac radioablation, also known as STAR (stereotactic arrhythmia radioablation) is an emerging treatment method for persistent ventricular tachycardia. Since its first application in 2012 in Stanford Cancer Institute, and a year later in University Hospital Ostrava, Czech Republic, the authors from all around the world have published case reports and case series, and several prospective trials were established. In this article, we would like to discuss the available clinical evidence, analyze the potentially clinically relevant differences in methodology, and address some of the unique challenges that come with this treatment method.

## 1. Introduction

Ventricular arrhythmias, including ventricular tachycardias (VT), are a major cause of sudden cardiac death (SCD) [1], which globally contributes to 4.25 million deaths every year [2].

The guidelines-driven management of VTs consists of antiarrhythmic drugs (AADs), placement of implantable cardiac defibrillators (ICDs), and the ablation of the arrhythmogenic substrate [3,4,5]. Despite the significant clinical improvement, results of conventional radiofrequency ablation (RFA) strategy can be limited especially in case of challenging anatomy and subepicardial substrate location. Post-RFA recurrence rate ranges from 12–17% at 1 year follow-up [6], and, although rarely directly associated with procedural complications (0.6%), RFA is associated with up to ~5% short-term mortality in ischemic VT cases [7].

A need to deliver better care for patients makes VT ablations one of the most dynamically growing areas in the field of cardiac electrophysiology (EP) [8]. The increasing number of patients with refractory VT leveraged by the implementation of new technologies has already improved clinical results [9]. Specifically, stereotactic arrhythmia radioablation (STAR) aims to reduce the burden of arrhythmia [10,11,12] through a combination of 3D electrophysiological mapping (EAM), noninvasive myocardial scar imaging (computed tomography (CT), positron emission tomography (PET), cardiac magnetic resonance (cMR) [13,14]), and noninvasive delivery of ablative radiation doses [15], as presented in Figure 1. STAR faces everyday clinical challenges associated with the extension of indications for VT ablations, including advanced ischemic and nonischemic cardiomyopathies [16,17,18]. As a routine step of the STAR methodology, cardiac imaging showed that a substantial number of VT patients have fibrotic scars located intramurally and epicardially, thus not being accessible with endocardial ablation approaches [19]. New insights confirm the necessity to co-register anatomical and functional data reassuring integration of imaging techniques with electrophysiological studies. Already, Andreu et al. [20] and Gupta et al. [21] showed favorable results of the cMR-EAM fusion for the ablation of VT substrate.

Last but not least, the complex management of VT patients is based on a holistic approach where ablation should be seen as one of many building blocks. It is important to note that a final clinical outcome reflects adequate selection of patients, procedural planning, and treatment strategy, as well as control of comorbidities (i.e., coronary artery disease, heart failure, and endocrine disorders) and aftercare [22]. As there is no “good fit for all” solution, personalized management pathways are essential.

The objective of this article is to familiarize the reader with the concept of STAR and review the available clinical data on its clinical applications. Most importantly, we would like to present a novel discussion on the possible reasons for treatment failures, significant differences in methodology, and finally suggest possible solutions based on the authors’ own clinical experiences.

## 2. Materials & Methods

The systematic literature review (Figure 2) was performed in November 2020 through the United States National Library of Medicine PubMed/Medline database (https://www.ncbi.nlm.nih.gov/; accessed on 18 February 2021). The search terms consisted of “ventricular tachycardia” or “VT” in combination with at least one of the following: “STAR”, “SBRT”, “stereotactic body radiation therapy”, “SRS”, “radiosurgery”, or “radioablation”. The initial search yielded 124 results. The first two authors selected publications based on abstract screening, the relevance criterion was defined as original full-text articles describing the treatment of at least one patient with stereotactic radiotherapy for arrhythmia. There were no significant disagreements. Then, the authors performed a secondary review of the bibliographies of the original articles and reviewed articles which referenced the selected papers. Finally, a total of 17 original articles regarding clinical applications of STAR were referenced in the review, including 11 case reports [23,24,25,26,27,28,29,30,31,32,33], 4 case series [10,12,34,35], and 2 clinical trial reports [11,36].

Data regarding ongoing and completed clinical trials were extracted from the United States National Library of Medicine Clinical Trial Register (https://clinicaltrials.gov/; accessed on 18 February 2021) using the same search terms as listed above for PubMed literature research. Eight studies were found in the database, two studies were included through secondary literature research, and another one was the authors’ contribution. Finally, 11 clinical trials are listed in the article, including the two whose results have already been published.

## 3. Results

### 3.1. Technical Considerations

For the majority of radiotherapeutic history, the collaboration with cardiologists mainly focused on the subject of damage control of radiation sequelae, including late (i.e., coronary disease) and early (i.e., ICD damage, thrombosis) toxicity. The introduction of curative radiosurgery into a strictly cardiological field caused a major paradigm shift and forced the development of many new and innovative strategies.

One of the major challenges in STAR was developing a reliable target delineation method, i.e., defining the volume to be irradiated on a 3D computer tomography reconstruction. This part, which is crucial for the treatment success, mainly relies on EAM, CT, and MR. However, the target delineation workflow is significantly different between studies and often based on indirect comparison of such data. As the arrhythmogenic substrate is not always well represented by medical imaging [37], we cannot exclude that some of the treatment failures reported in the literature were due to suboptimal target delineation.

Fortunately, many authors decided to challenge this problem. For example, in May and July of 2020, Brett et al. [38] and Hohmann et al. [39] published workflow descriptions on converting catheter-based EP maps to DICOM. The software developed by Hohmann et al. supports CARTO 3 (Biosense Webster), EnSite Velocity/Precision (Abbott), and RHYTMIA HDx (Boston Scientific) cardiac mapping systems. The solution published by Brett et al. uses proprietary MATLAB software and is compatible with CARTO 3 system. Dedicated software allows for contouring on an offline fusion of electroanatomic data with CT-reconstructed 3D models, which can later be imported to the radiotherapy planning system, allowing for higher precision than an indirect comparison between (CT/MRI-fused) EAM and treatment planning CT.

Regarding the optimal choice of treatment machine, both C-arm and CyberKnife^TM^ (CK) linear accelerators have been used in practice, and both are capable of delivering clinically acceptable dose distributions [40]. Moreover, recently the first patient has been successfully treated with an MR-linac [33]. The technical considerations of STAR are available in a thorough technical review recently published by Lydiard et al., and therefore will not be discussed here [41].

### 3.2. Mechanisms of STAR

The concept of applying stereotactic radiosurgery for the treatment of VT is generally attributed to Thomas Fogarty and Cyber Heart Inc. The early preclinical works suggested that a single dose of 25–35 Gy is capable of inducing fibrosis in the heart of a Göttingen minipig, which leads to a blockade of electrical activity in the myocardium [42,43]. Consequently, it was assumed that such a treatment protocol could be used in the treatment of VT.

This concept, however, has been challenged by other authors. Myocardial fibrosis is one of the late radiation effects, usually observed not sooner than at the end of the second month after irradiation, while the antiarrhythmic effect can occur almost immediately [44]. We can speculate that the answer lies in the difference between the radiobiology and dynamics of radiation-induced injury of the healthy heart tissue of a minipig and that of a myocardial scar in the human heart or simply in a different actual mechanism of the antiarrhythmic effect. For example, inflammatory cells are visible in the myocardium as soon as a couple of hours after the irradiation [45]. A recent study of four explanted hearts post-STAR by Kiani et al. [46] showed that the prevalent pattern within the irradiated region includes subendocardial necrosis surrounded by a rim of fibrosis and vascular changes. Despite the complicated and not yet well explained underlying mechanisms, STAR has already shown some promising clinical results.

### 3.3. Clinical Data—From the First Case Report to Clinical Series and Prospective Trials

So far, several case reports, case series, and prospective studies have been published in the literature. There is, however, a vast heterogeneity in terms of radiotherapy techniques and planning, patient selection, follow-up (FU), and outcome reporting. Below, we present full-text articles regarding the clinical applications of the stereotactic radiosurgery in VT treatment. The studies are described chronologically in Table 1 and Table 2.

The first full-text case report has been published by Cvek et al. in 2014 [23], presenting 72-year-old female treated with STAR for recurrent VT and arrhythmic storm (AS). No signs of toxicity or malignant arrhythmia were observed within the 120 days. A year later, Loo et al. [24] described a 71-year-old man treated in October 2012, the first in-human application of STAR. The treatment led to a reduction of VT, which lasted for 7 months until clinical recurrence, likely within the STAR target. The patient died at 9 months due to respiratory failure caused by chronic obstructive pulmonary disease exacerbation. The authors suggested that the RT dose might have been insufficient [47].

Two years later, Cuculich et al. [10] published a case series of five patients treated with STAR including a well-described and illustrated procedural workflow as supplementary material. The authors reported a 99.9% decrease in VT-burden after a 6 week blanking period. At 12 months, out of four patients alive, three were still no longer receiving AADs. One of the patients had to restart amiodarone at 9 months. Another patient had to receive catheter ablation at 4 weeks due to incomplete VT cessation, despite four prior ineffective catheter ablations. One patient died of a stroke 3 weeks after the treatment. Considering the significant disease burned, the association between this death and STAR remains unclear.

Finally, the results of the first prospective trial ENCORE-VT were published by Robinson et al. in 2018, shedding new light on the future of STAR [11]. The study included 19 patients, the majority presenting New York Heart Association Functional Classification (NYHA class) of III/IV (73.7%). The main endpoints were safety defined as up to 90 days with ≤20% treatment-related serious adverse event occurrence (CTCAE v4.0 grade III toxicity requiring hospitalization or any grade IV–V toxicity), which was met in 17/19 patients (89.5%). The efficacy was aimed for at least ≥40% of patients with any decrease of VT-burden in 6 months following treatment, (except a 6 week blanking period), and was achieved in 17 (94%) of the 18 evaluable patients. The two treatment-related serious adverse events (SAE) described in the article were a grade III heart failure exacerbation, which led to hospitalization 65 days after STAR, and grade III pericarditis, which led to hospitalization at 80 days and subsided after treatment. One of the patients died 17 days after the treatment due to an unrelated accident. The 50% and 95% reductions in VT episodes or 24 h premature ventricular contraction burden were achieved in 94% and 61% of the patients, respectively. However, as much as 69% experienced some VT recurrence between the end of the 6 weeks blanking period and 6 months after treatment. The authors made another important contribution a year later through a presentation during the ASTRO 2019 conference, where they discussed the late results of the trial [48], showing that perhaps the short FU could have hidden possible treatment-related sequelae. The 12 and 24 month survival rates were 72% and 58%, respectively. Among the eight deaths, one was unrelated (as described above), three unlikely (including an accident, amiodarone toxicity, and VT-recurrence), and finally four possibly related deaths, including two heart failures and two VT recurrences. Additionally, two late (>6 months) SAE occurred, including grade III pericardial effusion at 2.2 years and grade IV gastropericardial fistula at 2.4 years.

A month later, Neuwirth et al. [12] published an interesting case series consisting of 10 patients presenting with NYHA class II–III. At a median follow-up of 28 months (longest reported so far), only one patient experienced possible treatment-related grade III toxicity—a progression of mitral regurgitation. Three patients suffered non-arrhythmic deaths and were not listed as SAE. However, considering that the deaths at 43 and 54 months were due to heart failure, it cannot be excluded that the treatment influenced the underlying heart disease. During the follow-up, the VT burden was reduced by 87.5%, but the VT eventually recurred in 8 out of 10 patients, including two patients with increased VT burden compared to baseline.

In late 2019, Lloyd et al. [34] published a case series of 10 patients treated with STAR. Two of them were transferred to hospice shortly after STAR due to limited further treatment options and excluded from the VT-burden analysis. Out of the eight evaluable patients, seven responded to the treatment, which yielded a 69% reduction in all detected VT-seconds or 94% reduction after excluding the nonresponder. The authors reported that one of the patients experienced slow VT below the treatment zone during STAR and had to be resuscitated. However, based on the previously discussed reports, the relatively low toxicity can be associated with short follow-up (median—6 months), as many serious adverse effects develop late after STAR.

Eventually, a phase II clinical trial published by Gianni et al. [36] concluded that the efficacy was insufficient in terms of effective long-term arrhythmia control. The study included five patients presenting NYHA class II. Although no significant acute or early toxicity was reported by the authors, two of the patients died due to heart failure exacerbation during the follow-up. Most importantly, each of the five patients treated with STAR had clinically significant VT recurrence within the 14 months requiring prior doses of antiarrhythmic drugs.

Similarly, a recent case series by Chin et al. [38] reported limited efficacy of STAR. The study describes eight patients suffering from significant comorbidities (NYHA class III-IV), some of which were treated with doses lower than 25 Gy. The 3 month reduction in VT episodes was not statistically significant (*p* = 0.24). The authors, however, indicated that three of the patients (33%) experienced “apparent clinical benefit”. Each of them had NYHA class III, and two had no prior ablation (those were the only patients in this group without prior ablation). The third patient had an LVEF of 30%, the second-highest among the patients in the study group. Besides treatment planning aspects discussed later, the limited efficacy could have been associated with the severity of comorbidities in the non-responders.

### 3.4. Case Reports

Case reports, which are presented below and can be found summarized in Table 2, present interesting and innovative applications of STAR. Jumeau et al. [26] presented the first case of an intensive care patient treated for incessant VT as a rescue procedure. The treatment resulted in cessation of VT episodes during the 4 month FU and most importantly, an improvement in patient condition, which led to discharge from the hospital after 2 months of observation. Scholz [28] applied STAR for ventricular fibrillation of ischemic etiology, which could not be managed with standard treatment methods. The patient had complete cessation of arrhythmic episodes within 2 weeks and remained arrhythmia-free for the duration of the 60 day FU. Bhaskaran et al. [31] describe a case of STAR in a palliative setting due to advanced, metastatic neuroendocrine tumor, and life expectancy of months. The patient received standard catheter ablation but with limited effect, followed shortly by STAR which led to complete cessation of VT episodes in 6 days and no recurrence within the 60 day FU.

Other case reports include a successful treatment of VT originating from the right ventricle by Marti-Almor et al. [30]. The authors claim to have performed the first noninvasive STAR outside of the United States; however, to our best knowledge, the first STAR treatment in Europe should be attributed to Cvek et al. who published a case report five years earlier. Krug et al. [32] reported a case of partial response to STAR and death at 57 days due to sepsis-related cardiac failure. Finally, Mayinger et al. [33] recently published a case report on the first MR-guided STAR treatment. The patient remained free of VT episodes requiring ICD interventions up to the 3rd month after the treatment.

### 3.5. STARting to Think outside of the Box

Several of the case reports presented in Table 2 describe exciting applications of STAR outside of the usual cardiomyopathy setting. Haskova et al. [27] used STAR to treat VT caused by an unresectable cardiac fibroma. Although the treatment consisted in fact of a combination of surgical cryoablation, catheter ablation, and STAR, it effectively resulted in a gradual decrease of VT episodes during the 8 month follow-up despite a large, unresectable tumor. Zeng et al. [29] reported a case of VT originating from an intramural ventricular cardiac lymphoma. With the usage of a modified fractionation scheme (24 Gy in 3 fractions), the authors obtained a 4 month observation period free from VT. Monroy et al. [49] describe an interesting application of STAR in the treatment of atrial fibrillation through robotic radiosurgical pulmonary vein isolation due to 7 year history of paroxysmal atrial fibrillation. The treatment was effective for 6 months, after which permanent atrial fibrillation developed and the patient was further treated with pharmacotherapy. The most recent case report by Park et al. [25] describes a life-saving scenario in which STAR was applied for apical hypertrophic cardiomyopathy at a burnout stage. The patient presented hemodynamically unstable sustained monomorphic VT, and the radiofrequency catheter ablation was unsuccessful due to the induction of VF. Stereotactic delivery of 24 Gy in three fractions to the site of apical aneurysm resulted in an antiarrhythmic effect which lasted up to 6 months, without significant treatment-related toxicity.

### 3.6. Ongoing Clinical Trials

So far, two prospective clinical trials from the US are completed, published results, and can be found in Table 3 together with the ongoing trials. One of the Czech trials (NCT03819504) has changed its status to withdrawn, and another large study under the identifier of NCT04612140 has been published in October 2020. To the best of our knowledge, the third Czech trial (NCT03601832) should be soon approaching the end of recruitment. The German RAVENTA trial has been active for a year, and the description of the study protocol along with a short literature review has been published by Blanck et al. [50]. Four more clinical trials have recently opened recruitment in Italy, Japan, and two in Canada. Furthermore, a Japanese trial regarding the application of STAR in atrial fibrillation with a clinical trial identifier of UMIN000031322 has finished recruiting of three patients, and we are currently awaiting the publication of the results.

Last but not least, we would like to present the Polish SMART-VT trial, which has recently started recruiting and treated the first patient on the 10th of December 2020. We encourage both physicians and patients to contact the corresponding author regarding qualification for the treatment. The inclusion criteria can be found briefly described online (https://clinicaltrials.gov/ct2/show/NCT04642963; accessed on 18 February 2021).

## 4. Discussion

### 4.1. Target Volume Delineation

As mentioned earlier, the precise radiotherapy target delineation is crucial for the treatment success, but the variability in reported treatment volumes is significant, and planning treatment volumes (PTV) range from a mean of 22.15 [12] to 143 cc [36] between the published trials and case series. Moreover, the recent report from the RAVENTA trial [51] shows a concerning lack of reproducibility, including almost 14-fold volume difference (5.9 vs 79.9 cc) in clinical target volume (CTV) and conformity indices as low as 0.02 between identical cases contoured by five different university centers. To our best knowledge, however, the biggest differences lies in methodology.

Gianni et al. [36] reported the highest PTVs, with a mean value of 143 cc. The authors report that arrhythmia substrate was identified on a “contrast-enhanced first-pass CT” and correlated with the electroanatomic mapping from the previous ablation, as well as 12-lead ECG, and an additional 3 mm margin was added for the PTV. Chin et al. [35] used as much as 6–8 mm margins, yet the mean PTV was only 121.4 cc. Once again, the authors based their target volume definition on CT, combined with MRI, and the EP was reduced to 12 lead ECG and data from previous ablations, if available. Both of these studies reported disappointing results, with a 100% recurrence rate in Gianni’s series, and clinical benefit in only 33% of the patients reported by Chin.

Robinson et al. [52] and Lloyd et al. [34] opted for a different approach, which included both scar homogenization based on medical imaging, and arrhythmia substrate ablation based on pretreatment non-invasive (Robinson) or invasive (Lloyd) EAM. In the study by Robinson, the 5 mm margin for PTV resulted in a median of 98.9 cc, while Lloyd reports 1–5 mm margins and an average of 81.4 cc. As described earlier (Table 1), both of these studies showed a large number of responders and as much as 94% reduction in total VT-burden, after excluding the one non-responder in Lloyd’s study.

Finally, Neuwirth et al. [12] reduced the treatment volume to “critical part of the substrate” based on pre-existing EAM from previous partially or noneffective ablation, omitting the additional margins and inclusion of anatomical scar, which resulted in mean PTV as low as 22.2 cc. The authors claim that compared to Robinson, the lower PTV can be attributed to the different irradiation technique (CK instead of C-arm). The so-called gross treatment volume was not that different in Robinson’s study (median 25.4 cc). On the other hand, the highest PTV among all studies (almost six times higher), was reported in the second CK series [36]. The authors achieved an 87.5% decrease in VT burden but as much as 80% recurrence after the blanking period.

Based on this data, we believe that at the moment the EAM should be regarded as pivotal for STAR, and unnecessary large treatment volumes should be avoided, as data suggests that relatively small irradiated volumes might be sufficient for VT reduction.

### 4.2. C-Arm or Cyber Knife?

In the article by Gianni et al. [36], we can find that the authors used “X-Sight Spine Tracking System” and “Synchrony Respiratory Tracking System”, which tracked the ‘trans-jugular temporary pacing lead’ implanted before CT. It means that the positioning was automatically adjusted to the patient’s spine through orthogonal X-rays performed pre-and during the treatment, and the respiratory movement was compensated by continuous robotic arm movement based on temporary lead position and chest wall motion. In other words, the positioning of the treatment volume was assessed only indirectly, and it was not possible to visualize the heart on the treatment couch due to the lack of cone beam computed tomography (CBCT). Neuwirth et al. used a significantly different approach and targeted the ICD-lead only, both for positioning and respiratory tracking. It seems more logical as the lead is fixed in the treated organ. However, to achieve optimal fiducial tracking, at least three reference points would be necessary for triangulation. From our experience, the ICD lead can serve as one, sometimes two, hence the risk of unaccounted geometrical shifts cannot be excluded. Moreover, Wang et al. suggested that ICD lead might be suboptimal for tracking due to artifacts from the right ventricular coil [40].

The remaining “C-arm” authors have all used 4D-CT to account for respiratory motion and CBCT-based positioning. This approach allows for a careful revision of patients’ anatomy on the treatment table and precise correction of positioning based both on the bony structures and heart but lacks the automatic positioning adjustments during treatment available on CK.

Based on our experience, we use the C-arm approach with the addition of either deep inspiration breath hold (*n* = 4) or free breathing respiratorygating (*n* = 1), depending on the patient’s compliance. The CBCT imaging allowed for the mitigation of one possible serious adverse effect so far, in a patient who drank approximately 300 mL of dense liquid before the treatment despite clear physician’s instructions, which caused the stomach to expand dangerously close to the irradiated volume. Moreover, we have found that despite significant artifacts caused by the ICD lead, the positioning of the target volume and cardiac subvolumes is facilitated through the identification of reference points, such as calcified atherosclerotic plaques, abundantly found in the STAR patients population (Figure 3).

### 4.3. The Risk Profile—Early Toxicity

There has been little-to-no concern regarding the early toxicity in the literature, and the majority of the adverse effects reported are grade I or II, often subclinical. However, considering the significant variability in toxicity reporting, it is difficult to form general conclusions. For example, Robinson et al. [11] reported common occurrence of fatigue, hypotension, dizziness, dyspnea, and nausea. Moreover, radiation pneumonitis (11.1%), pericardial effusions (28%) including one case of grade III SAE, and a grade III heart failure exacerbation were also reported. On the other hand, Gianni et al. [36] found no acute or early “radiation complications” despite the highest irradiated PTVs, while Neuwirth et al. [12], using the lowest PTVs, observed only nausea in 40% of the patients. Nevertheless, the risk of clinically significant early toxicity seems low, and except for one case of slow VT below the treatment zone requiring resuscitation described by Lloyd et al. [34], no significant periprocedural toxicity has been observed either.

### 4.4. Late Toxicity

So far, only Neuwirth et al. [12] and Robinson et al. [48] have presented results with 2+ years of FU. The authors reported three probably or definitely treatment-related SAE, including grade 3 progression of mitral regurgitation at 1.4 years, grade 3 pericardial effusion at 2.2 years, grade 4 pericardial fistula at 2.4 years, adding up to a total of three cases of significant late treatment toxicity in 29 patients. Considering that none of the other 15 studies report on a median FU longer than 12 months, we would like to voice our concern that the risk of significant late adverse effects can be significantly underestimated, especially considering that the authors using the largest PTVs reported only 12 and 7.8 months median FU, respectively [35,36]. Moreover, a randomized clinical trial is necessary to assess the impact of STAR on the risk of cardiac failure exacerbation, which has been reported as a cause of mortality by the authors.

## 5. Conclusions

The future of STAR is still undetermined. Despite many promising results, the methodology of this emerging treatment method is significantly different between studies, including a significant lack of agreement in terms of the treatment volumes. Despite the favorable early toxicity profile, studies have shown clinically relevant late toxicities at significantly later timepoints than the follow-up available in the majority of the reports. Nevertheless, considering a significant amount of promising data, we believe that the development of optimal STAR workflow is possible and could help to address a specific and clinically important problem of the postablation VT recurrences.

## Figures and Tables

**Figure 1 jcm-10-01238-f001:**
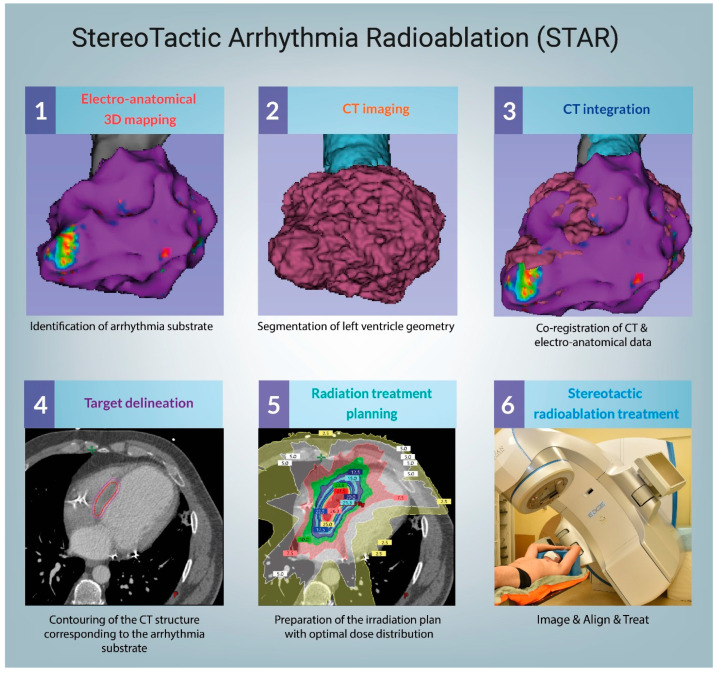
Procedural workflow. CT–computed tomography.

**Figure 2 jcm-10-01238-f002:**
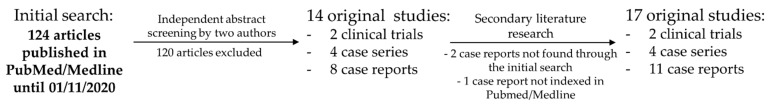
Literature research workflow.

**Figure 3 jcm-10-01238-f003:**
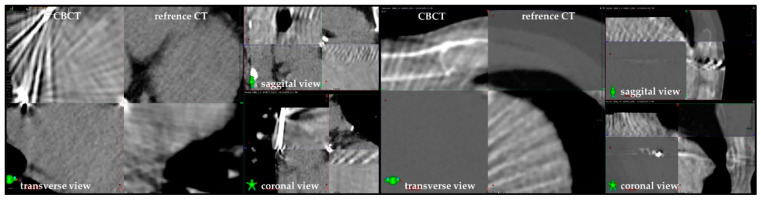
Pre-treatment cone beam CT (**upper**-**left** and **lower**-**right** parts) fused with treatment planning CTs. High-density reference points facilitate precise image-guided radiotherapy, despite significant artifacts caused by the Implantable Cardioverter Defibrillator lead.

**Table 1 jcm-10-01238-t001:** Clinical data on stereotactic arrhythmia radioablation (STAR).

Date	1st Author; Type	#, Irradiation Method	Median Age	Etiology; Mean LVEF (%)	Median Follow-Up (Months)	Description of Treatment Outcome and Toxicity
July 2014	Cvek [23]; Case report	1; CK	72	Dilated cardiomyopathy; 25%	4	- No episode of malignant arrhythmia for 4 months
						- No signs of toxicity
June 2015	Loo [24]; Case report	1; CK	71	Ischemic; 24%	9	- Frequent nonsustained and pace-terminated VT occurred 3 months post-STAR after reducing the dose of sotalol and mexiletine
						- Recurrent VT and COPD exacerbation at 9 months after STAR followed by death
December 2017	Cuculich [10]; Case series	5; C-arm	62	Mostly non-ischemic (60%); 23%	12	- VT episodes decreased by 99.9% after a 6 week blanking period in all patients
						- No clinically significant STAR-related adverse effects. One patient died of a stroke at 3 weeks after treatment, unclear association with STAR
November 2018	Robinson [11]; Clinical trial	19; C-arm	66	Mostly ischemic (58%); 25% ^	13	- 50% and 95% reduction in VT episodes or 24 h PVC burden in 94% and 61% of the patients, respectively
						- Two grade III treatment-related SAE (heart failure exacerbation, pericarditis), no grade 4 toxicity
December 2018	Neuwirth [12]; Case series	10; CK	64	Mostly ischemic (80%); 26.5%	28	- VT burden decreased by 87.5% over 28 months
						- Mild toxicity, one case of possibly related grade III toxicity within follow-up period—gradually progressing mitral regurgitation
September 2019	Lloyd [34]; Case series	10; C-arm	61	Mostly nonischemic (60%); N/A	6 *	- 69% reduction of VT burden in evaluable patients (8/10) within 176 day, 94% reduction after excluding the single non-responder (7/10)
						- Two patients with mild clinical and radiographic signs of pneumonitis responsive to steroid therapy, one patient resuscitated due to VT during STAR treatment
March 2020	Gianni [36]; Clinical trial	5; CK	61	Mostly ischemic (80%); 34%	12 *	- Clinically significant VT recurrence in all patients
						- No STAR-related acute or early radiation complications, however, two of the patients died of heart failure exacerbation at 10 and 12 months
August 2020	Chin [35]; Case series	8; C-arm	74	Ischemic/nonischemic (even); 21%	7.8	- No statistically significant difference between the total number of ICD therapies (VT episodes, ATP/ICD shocks) recorded 3-month pre- and post-STAR. “Apparent clinical benefit” was observed in 33% of the patients.
						- No acute periprocedural complications. Two non-STAR-related deaths at 2 months, one unclear (multiple ICD shock till 6th week, opted out of ICD therapy).

#—number of cases; LVEF—left ventricle ejection fraction; ATP—antitachycardia pacing; ICD—implantable cardioverter–defibrillator; STAR—stereotactic arrhythmia radioablation; VT—ventricular tachycardia; CK—CyberKnife. ^—median *—mean.

**Table 2 jcm-10-01238-t002:** Case reports.

Date	1st Author	Age; VT Type	Etiology	Follow-Up	Irradiation Method	Description of Treatment Outcome and Toxicity
May 2018	Jumeau [26]	75; Incessant VT (polymorphic)	Severe dilated cardiomyopathy	4 months	CK	Free from VT up to 4th month after STAR
October 2018	Haskova [27]	34; Recurrent VT (monomorphic)	Cardiac fibroma	8 months	CK	VT gradually subsided within 8 months after STAR
March 2019	Scholz [28]	53; Ventricular fibrillation	Ischemic	60 days	C-arm	Cessation of arrhythmic episodes in 2 weeks and no recurrence within 60 days
June 2019	Zeng [29]	29; Recurrent VT (two morphologies)	Cardiac Lymphoma	4 months	CK	Free from VT up to 4th month after STAR
July 2019	Marti-Almor [30]	64; Incessant VT (monomorphic)	Right ventricular cardiomyopathy	4 months	C-arm	Free from VT up to 4th month after STAR
September 2019	Bhaskaran [31]	34; VT storm	Unknown	60 days	C-arm	Cessation of arrhythmic episodes in 6 days and no recurrence within 60 days
Octobr 2019	Krug [32]	78; Recurrent VT (monomorphic)	Dilated cardiomyopathy	57 days	C-arm	Partial VT burden reduction. The patient developed sepsis-associated cardiac circulatory failure which led to death 57 days after treatment
February 2020	Mayinger [33]	71; Recurrent VT (polymorphic)	Nonischemic	3 months	MR-linac	Immediate aggravation of the clinical VT (48 h) followed by cessation of VT for the rest of the FU
July 2020	Park [25]	76; Recurrent VT (monomorphic)	apical hypertrophic cardiomyopathy	6 months	C-arm	Despite two occurrences of sustained VT at 6 and 8 weeks, patient remained free from ICD shocks up to 6th month after STAR.

ATP—antitachycardia pacing; ICD—implantable cardioverter–defibrillator; STAR—stereotactic arrhythmia radioablation; VT—ventricular tachycardia; CK—CyberKnife.

**Table 3 jcm-10-01238-t003:** Ongoing clinical trials.

Date of Start	Full Name of the Trial	Country of Origin	Clinical Trial Identifier	Status	Planned Number of Participants
February 2015	CyberHeart’s Cardiac Arrhythmia Ablation Treatment: Patients with Refractory Ventricular Tachycardia	USA	NCT02661048	Active, not recruiting	5
				Primary objectives completed	
				Results published	
July 2016	Phase I/II Study of EP-guided Noninvasive Cardiac Radioablation for Treatment of Ventricular Tachycardia	USA	NCT02919618	Active, not recruiting	19
				Primary objectives completed	
				Results published	
August 2018	Phase I/II Study of 4-D Navigated NonInvasive Radiosurgical Ablation of Ventricular Tachycardia	Czech Republic	NCT03601832	Active, recruiting	10
August 2018	STereotactic Ablative Radiosurgery of Recurrent Ventricular Tachycardia in Structural Heart Disease	Czech Republic	NCT03819504	Withdrawn	50
September 2019	STereotactic RadioAblation by Multimodal Imaging for VT	Italy	NCT04066517	Active, recruiting	15
September 2019	Minimally Invasive Arrhythmia Treatment with External Radiation Therapy for Intractable Ventricular Tachycardia	Japan	jRCTs032190041	Active, recruiting	3
December 2019	Radiosurgery for the Treatment of Refractory Ventricular Extrasystoles and Tachycardias	Germany	NCT03867747	Active, recruiting	20
January 2020	Stereotactic Arrhythmia Radioablation for Ventricular Tachycardia Management	Canada	NCT04065802	Active, recruiting	20
August 2020	Cohort Study—SBRT for VT Radioablation	Canada	NCT04162171	Not yet recruiting	12
October 2020	STereotactic Ablative Radiosurgery of Recurrent Ventricular Tachycardia in Structural Heart Disease	Czech Republic	NCT04612140	Active, recruiting	100
September 2020	Stereotactic Management of Arrhythmia—Radiosurgery in Treatment of Ventricular Tachycardia	Poland	NCT04642963	Active, recruiting	11

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
