# Peer review of "Clinical Evidence behind Stereotactic Radiotherapy for the Treatment of Ventricular Tachycardia (STAR)—A Comprehensive Review"

_jcm, 2021, doi:10.3390/jcm10061238_

Round 1
Reviewer 1 Report
Dear Authors,
I have read the manuscript entitled “ Clinical evidence behind stereotactic radiotherapy for the treat-ment of ventricular tachycardia (STAR)—a comprehensive re-view.”
The subject of work is interesting and manuscript is carefully prepared.
I do not make any comments and recommend publishing the manuscript in the present form.
Kind Regards
Author Response
Dear Reviewer,
Thank You for your kind review, we are very pleased to see such positive receipt of our article, and hope that many readers will find it similarly interesting in the future.
Reviewer 2 Report
This is a very interesting review that provides a synthesis of current knowledge on Stereotactic arrhythmia radioablation (STAR) as an emerging treatment method for persistent ventricular tachycardia after unsuccessful radiofrequency ablation. The article discusses the available evidence in this field, taking into account uncertainty and still limited data. The writing is clear and the structure satisfying.
Major issues
-There are no major flaws in the paper.
Minor Issues
- Introduction/page 2/first paragraph: “…Unfortunately, the recurrence rate after VT substrate ablation during sinus rhythm ranges from 12-17% at 1 year follow-up [18], and radiofrequency ablation procedures of scar-related VTs are associated with ~5% short-term mortality, especially in patients with peri-procedural complications, including thrombotic events. “. As it is drafted, it would suggest that the 5% short-term mortality from radiofrequency ablation is almost entirely due to procedural complications. These patients are in a severe condition with high risk of mortality. According with the citation, refractory VT is the cause of death in 22% of cases. In a 39% the cause of death is cardiac but not related to the VT (most commonly advanced heart failure) and another 11% die of non-cardiac events. Only 0,6% is related to a major procedure complication. It would be important to expand this information to clarify that radiofrequency ablation is the first-line therapy in recurrent VT and when it is performed in experienced centers has a reasonable success rate and low risk of serious complications.
- Results/page 4/Second paragraph: “…Electrophysiologists, on the other hand, use a catheter to directly map the arrhythmogenic substrate, and are not necessarily accustomed to working with CT or MR.” This statement does not conform to reality. The use of MRI and CT images is quite frequent in electrophysiology laboratories. It is a common practice to integrate these images with those obtained with electroanatomical mapping during the ablation procedure. This allows achieving more accurate reconstructions and improving precision. Also, in recent years late gadolinium enhancement cardiac magnetic resonance and multidetector cardiac computed tomography has emerged as novel imaging tools for arrhythmogenic substrate identification and characterization. Using pixel signal intensity maps or wall thickness maps delivered from MRI or CT, respectively, electrophysiologist could decide the best procedure planning and approach; complete substrate identification and characterization and focus electroanatomical mapping in regions of interest. I recommend the authors to read the article: Berruezo A, Penela D, Jáuregui B, Soto-Iglesias D. The role of imaging in catheter ablation of ventricular arrhythmias. Pacing Clin Electrophysiol. 2021 Feb 1. doi: 10.1111/pace.14183. Epub ahead of print. PMID: 33527461.
Author Response
Dear Reviewer,
First of all, we would like to thank you for your careful review and important comments. We have addressed both issues which you pointed out hoping you will find the responses and revised version of the manuscript satisfactory., The editions and changes in the text are marked in red.
- It is true that such presentation of data could lead the reader to incorrect conclusions and we would like to clarify that RFA is a first-line treatment for VT ablations, while SMART can be considered only in patients with prior ineffective RFA or contraindications.. Moreover, we would like to inform the readers that the number of complex VT cases is growing. Thus, a personalized treatment strategies are essential in clinical practice and STAR might be a treatment of choice in a selected groups of patients. Therefore, we decided to modify the existing sentence according to your suggestions, remove unnecessary possible bias against RFA, but most importantly add additional paragraphs addressing the current state & limitations of RFA, which can be now found on page 2, highlighted in red.
- Thank you for this very important feedback. We agree that the previous statement was inaccurate. The information regarding increasing use of medical imaging in ablation procedures was included a section in the introduction paragraph focusing on the application of imaging techniques in the VT ablation planning and treatment. The previous sentence was perhaps too much of an opinion based on personal experience which cannot intersect with evidence-based information presented in the review article. Moreover, the radiotherapy-electrophysiology workflow might be an interesting subject for the readers. Saying that, we have elaborated on the procedural step-by-step approach. In our experience, the initial target delineation combines results of the 3D electroanatomicla mapping (EAM) and pre-procedural computed tomography (CT) imaging of the heart prepared simultaneously with the EPM, using a mobile contouring station and screen-to-screen method. Subsequently, we apply the post-hoc image integration strategy described by Hohmann et al. allowing for a co-registration of EAM and CT data using EP Maping plugin for the Slicer3D software. Our preliminary observations show approximately 80-90% concordance between the aforementioned methods of target volume definition. On contrary, some of the authors show significant differences. We addressed this subject in the ‘Target volume delineation’ paragraph on page 10.
Hohmann S, Henkenberens C, Zormpas C, et al. A novel open-source software-based high-precision workflow for target definition in cardiac radioablation. J Cardiovasc Electrophysiol. Epub ahead of print 2020. DOI: 10.1111/jce.14660.
Reviewer 3 Report
Major comments:
- While the literature clearly shows data about low success rates associated with RF ablation of VT substrate, what are the possible causes of this limited low success? I mean, I think that these causes could be the ones that motivated the development of new ablative or guidance techniques. At this regard, what can RT contribute to RF? Authors say “Due to the aforementioned limitations” but they really talked about success rates, not about identified specific limitations.
- I suggest including the flow chart of the searching of clinical data in Pubmed, indicating number of results, number of excluded and reasons, etc. And explain the used relevance criterion. Were other resources, e.g. Cochrane Database, considered? Consider in the future to include other information source: MAUDE, US database reporting manufacturer and user facility device experience (adverse events, malfunctioning, etc.)
- While the authors have completed an excellent review of available clinical data about the topic, I miss in the conclusions a comment about the current risk-benefit profile from all the data analyzed. In other words, a sort of executive summary that briefly deals with the residual risks that still remain despite the clinical evidence of safety shown by the clinical data. I suggest the authors try to take some ideas from the clinical evaluation procedure described in the MEDDEV whitepaper (Clinical Evaluation. MEDDEV 2.7/1 Rev 4). Beyond the raw clinical data, it is interesting to read the author’s opinion about the current risks and benefits of this technology applied to treat cardiac arrhythmias.
Author Response
Dear Reviewer,
Thank You for your valuable thorough review. It is my strongest belief that your comments significantly improved the quality of the manuscript, and most importantly, allowed us to improve the clarity for the reader and bring forth conclusions that were previously lost in-between lines. Regarding the specific comments (changes in manuscript are marked in red):
1. We have added additional paragraphs in the introduction which address the specific limitations (such as localization & complex VT origins) of RF ablation, and the possible contribution of RT. Generally, we have also screened the article for other statements like ‘aforementioned limitations’, and made sure that each of them refer to specific explanations in the manuscript.
2. An additional explanation on the relevance criterion was added on page 3, and a graph describing the literature research workflow was included (graph 2 on page 4). To maintain consistency in graphical layout, we prepared a horizontal graph, but we will gladly change it to a vertical, PRISMA-like graph, if preferable. Unfortunately, we have not used other sources such as Cochrane Database for the ‘standardized’ literature research, and thus this remark is very accurate. We will include such in the future.
3.This comment was crucial for the manuscriptand we hope that re-designed paragraph helped to improve clarity and conclusions. The subject of the risk-benefit profile is a very complex matter, as in fact, from a radiation oncologist, the STAR is currently a huge basket comprising of significantly different treatment methods. Based on my own experience (visits & internships in two biggest STAR centers in EU, treating RO in SMART-VT trial and active member of STOPSTORM), I believe that the differences in target volume definition, and equally importantly positioning on the treatment couch, are significant and might as well be one of the reasons behind limited efficacy. Moreover, despite the somehow expected differences between treated volumes at different institutions (mean values ranging from 21 to 141 cc), the recent report from RAVENTA trial showed that a very significant variability can be observed even between different observers within one clinical trial. We have moved all the information regarding the aforementioned matters to the discussion, and elaborated on differences and their consequences. Moreover, we analyzed some of the previously unaddressed methodological differences, such as details of patients’ positioning. The new part of the manuscript can be found on pages 10-12. The last paragraph contains the risk-profile data, although limited by significant differences in methodology between studies.
Reviewer 4 Report
This is a manuscript of a topic of current interest. The authors aimed to review the current clinical evidence of stereotactic radiotherapy for the treatment of ventricular arrhythmias. The authors conclude that the current role and further potential of STAR is still undetermined. Important data, but several limitations exist with this manuscript, and these limit the generalizability and impact.
- The manuscript requires language editing
- Highlight to need of personalized paths in VT management and the role of MRI in terms of lesion visualization and assessment.
- The information provided by the authors is not new and a lot of data has already been published about STAR. So please emphasize what is really a novel information from your data.
- The technical considerations section requires revision. The paragraph about EP and electroanatomical-mapping is not appropriate.
- Consider to include more representative figures.
- Many parts of the article are redundant. You should focus on what is really relevant and consecutively shorten the manuscript.
- Use sub-headings in the discussion section.
- I miss information/ discussion about procedure related complications.
Author Response
Dear Reviewer,
Thank You for your critical review, we appreciate the thorough revision and tried our best to respond to each of the remarks as accurately as possible. Major changes are marked in red in the manuscript. Regarding the specific comments:
- The language has been now reviewed language-wise by an English native speaker, we identified many minor mistakes which were subsequently corrected.
- We have highlighted the importance of CT and MRI imagining in the introduction, and elaborated on the personalized treatment volume identification in the discussion (‘Target volume delineation’, page 10)
- This is a very valid point, as we have not emphasized enough the ‘clue’ of this review. Being aware of the ongoing scientific STAR-related discussion, we tried to discuss the matter from a clinician’s point of view, as we believe that the available articles and reviews offer little critical discussion regarding important practical aspects of the treatment. We have discussed significant differences between methodology used by authors, including contradicting statements (i.e., the use of ICD leads in CK-based STAR), invasive, non-invasive or optionally use of electroanatomic mapping. We’ve addressed the little agreement on the precision of STAR (PTV margins ranging from 0 to 8mm), significant differences between mean PTVs presented by authors (21 – 143cc), and finally as much as 14-fold volume differences in the same target delineated by different observers within one trial (RAVENTA, Boda-Heggemann). Finally, we provide the reader with solutions to these problems, such as the target delineation workflows provided by Brett and Hohmann. Nevertheless, all of this was somehow scattered within the manuscript text. We have gathered this data, and rewritten the discussion on practical treatment aspects, which can be now found on pages 10-12.
- According to this suggestion, which was also voiced by other reviewers, we have rewritten the ‘EP’ section in introduction paying close attention to highlighting the importance of medical imaging.
- We have added two more figures – literature search workflow and a practical aspects of cone-beam CT-based patient’s positioning.
- We have reorganized the article moving some of the data to dedicated paragraphs, and shortened the descriptions through language editing. Also, we have removed the unnecessary references to future studies (such as STOPSTORM). However, with consideration to other reviewer’s suggestions, we did not manage to significantly shorten the article. It is our sincerest belief that in the current version, less (or hopefully none) data will look redundant.
- We have reorganized the discussion and added subheadings.
- We have prepared a new paragraph at the very end of the discussion, which regards the procedural complications.
Boda-Heggemann J, Blanck O, Mehrhof F, et al. Interdisciplinary clinical target volume generation for cardiac radioablation: Multi-center benchmarking for the RAdiosurgery for VENtricular TAchycardia (RAVENTA) trial. Int J Radiat Oncol. Epub ahead of print 2021. DOI: 10.1016/j.ijrobp.2021.01.028.
Round 2
Reviewer 4 Report
This is a revised version of the initial manuscript. The paper improved. The authors aimed to review the current clinical evidence of stereotactic radiotherapy for the treatment of ventricular arrhythmias. Some limitations still exist with this manuscript.
- The introduction section is too long and not focussed enough. I suggest to focus on the real relevant publications instead of listing >30 citations.
- Text and tables in the results section are redundand. Shorten the text and refer to the corresponding tables in the text.
- Figure 3 is confusing and requires revision.
Author Response
Dear Reviewer,
Thank You for the valuable review. We have adjusted the manuscript according to your suggestions, and we believe that the clarity improved, and the article is more focused on important data now.
- We have greatly shortened the introduction, leaving the most valuable information and suggestions from other reviewers.
- Information such as age, LVEF and etiology were moved to the Table 1 altogether. Moreover, data present both in Table and manuscript was shortened, if possible. The manuscript is significantly shorter now.
- We have included more thorough description of the Figure 3. It refers to a very specific radiotherapy-related part of STAR, but it should be clearly understandable now for readers that are familiar with Image Guided RadioTherapy (IGRT).